# Fatal Puumala Hantavirus Infection in a Patient with Common Variable Immunodeficiency (CVID)

**DOI:** 10.3390/microorganisms11020283

**Published:** 2023-01-21

**Authors:** Philipp Steininger, Larissa Herbst, Karl Bihlmaier, Carsten Willam, Sixten Körper, Hubert Schrezenmeier, Harald Klüter, Frederick Pfister, Kerstin Amann, Sabrina Weiss, Detlev H. Krüger, Robert Zimmermann, Klaus Korn, Jörg Hofmann, Thomas Harrer

**Affiliations:** 1Institute of Clinical and Molecular Virology, Universitätsklinikum Erlangen, Friedrich-Alexander-Universität Erlangen-Nürnberg, 91054 Erlangen, Germany; 2Department of Nephrology and Hypertension, Universitätsklinikum Erlangen, Friedrich-Alexander-Universität Erlangen-Nürnberg, 91054 Erlangen, Germany; 3Institute for Clinical Transfusion Medicine and Immunogenetics Ulm, German Red Cross Blood Transfusion Service Baden-Württemberg-Hessen and University Hospital Ulm, 89081 Ulm, Germany; 4Institute of Transfusion Medicine, University of Ulm, 89081 Ulm, Germany; 5Institute of Transfusion Medicine and Immunology, German Red Cross Blood Transfusion Service Baden-Württemberg-Hessen, Medical Faculty Mannheim, Heidelberg University, 68167 Mannheim, Germany; 6Department of Nephropathology, Institute of Pathology, Universitätsklinikum Erlangen, Friedrich-Alexander-Universität Erlangen-Nürnberg, 91054 Erlangen, Germany; 7Institute of Virology, Charité-Universitätsmedizin Berlin, 10117 Berlin, Germany; 8Department of Transfusion Medicine and Hemostaseology, Universitätsklinikum Erlangen, Friedrich-Alexander-Universität Erlangen-Nürnberg, 91054 Erlangen, Germany; 9Infectious Disease and Immunodeficiency Section, Department of Internal Medicine 3, Universitätsklinikum Erlangen, Friedrich-Alexander-Universität Erlangen-Nürnberg, 91054 Erlangen, Germany

**Keywords:** Puumala hantavirus, encephalitis, common variable immunodeficiency, ribavirin, convalescent plasma

## Abstract

Puumala hantavirus (PUUV) infections usually show a mild or moderate clinical course, but may sometimes also lead to life-threatening disease. Here, we report on a 60-year-old female patient with common variable immunodeficiency (CVID) who developed a fatal PUUV infection with persistent renal failure, thrombocytopenia, and CNS infection with impaired consciousness and tetraparesis. Hantavirus-specific antibodies could not be detected due to the humoral immunodeficiency. Diagnosis and virological monitoring were based on the quantitative detection of PUUV RNA in blood, cerebrospinal fluid, bronchial lavage, and urine, where viral RNA was found over an unusually extended period of one month. Due to clinical deterioration and virus persistence, treatment with ribavirin was initiated. Additionally, fresh frozen plasma (FFP) from convalescent donors with a history of PUUV infection was administered. Despite viral clearance, the clinical condition of the patient did not improve and the patient died on day 81 of hospitalization. This case underlines the importance of the humoral immune response for the course of PUUV disease and illustrates the need for PCR-based virus diagnostics in those patients. Due to its potential antiviral activity, convalescent plasma should be considered in the therapy of severe hantavirus diseases.

## 1. Introduction

Hantavirus disease, also known as hemorrhagic fever with renal syndrome (HFRS) in Europe and Asia or hantavirus cardiopulmonary syndrome (HCPS) in America, is an emerging zoonosis with widespread distribution. Human pathogenic hantaviruses belong to the genus Orthohantavirus. They are carried by rodents that act as virus reservoirs and are transmitted to humans by the excreta of these natural hosts. Puumala hantavirus (PUUV) is the most prevalent hantavirus in Europe and is harbored by bank voles (*Myodes glareolus*). In contrast to infections by some other European hantavirus species which can cause severe hantavirus disease with case fatality rates (CFRs) of up to 12% [1], the clinical course of PUUV infections is usually mild or moderate with CFRs below 0.4% [2,3,4]. Common variable immunodeficiency (CVID) is the most frequent clinically relevant primary immunodeficiency and is primarily characterized by a hypogammaglobulinemia which leads to an increased risk for severe viral and non-viral infections [5].

Convalescent plasma of otherwise healthy donors who have recovered from hantavirus infection may contain protective anti-hantavirus antibodies. The data on convalescent plasma therapy, however, are limited. In a first trial of human convalescent plasma for treatment of HCPS caused by Andes hantavirus, a decrease in CFR with borderline significance was observed [6].

Here, we report on a case of fatal PUUV infection in a patient with CVID who was unable to generate a PUUV-specific antibody response. To our knowledge, this is also the first report on the treatment of a PUUV infection with convalescent plasma. This study has implications for laboratory diagnosis, treatment options, and pathogenesis.

## 2. Case Presentation

A 60-year-old female patient from a rural area in southwest Germany was admitted to a peripheral hospital with fever, headache, syncope, and altered consciousness (day 1). Due to the suspicion of community-acquired pneumonia in the chest radiography, an empiric antibiotic treatment with piperacillin/tazobactam and clarithromycin was initiated. Renal function was initially normal, but creatinine level increased within three days to 3.5 mg/dL. At the age of 34 years, she had been diagnosed with CVID with an absence of B cells in the peripheral blood and decreased levels of IgG, IgA, and IgM. Since then, she regularly received immunoglobulins, most recently a total of 12 g of Hizentra^®^ subcutaneously every week.

Due to the deterioration of her condition, she was transferred to our hospital on day 4. Laboratory analyses showed a normal leukocyte count (8.78/nL), moderate lymphocytopenia (0.9/nL), thrombocytopenia of 37/nL, and elevation of C-reactive protein (155.4 mg/L, reference value, <5 mg/L) and procalcitonin (5.53 ng/mL, reference value, <0.5 ng/mL). IgG levels in the serum was as low as 5.89 g/L (normal range, 6.1–16.2 g/L) and neither IgM nor IgA antibodies were detectable. Flow cytometric analysis of peripheral lymphocytes showed 2331 CD3+ T cells/µL (normal range, 600–2200 cells/µL) but no detectable CD19+ B cells. A CT scan on day 6 revealed a large pleural effusion and ascites but no signs of pulmonary infiltrates.

Within a few days the patient developed renal failure [creatinine 6.1 mg/dL (reference value < 1.1 mg/dL), urea 218 mg/dL (reference range, 17–43 mg/dL), eGFR 7 mL/min (reference value, >60 mL/min)] with anuria requiring hemodialysis. Urine analysis showed leukocyturia and proteinuria (14.7 g/L) without casts or cell cylinders. A kidney ultrasonography was found to be normal. On day 11, the patient developed decreased vigilance and symmetrical, proximal flaccid paralysis beginning at the upper extremities with progression distally and to the lower extremities. Magnetic resonance imaging of the brain (day 12) and spine (day 18) did not reveal any relevant pathologic results. Cerebrospinal fluid (CSF) analysis on day 12 showed no pleocytosis (3 leukocytes/µL) but highly elevated protein (3.3 g/L). She was first treated with intravenous immunoglobulins and then with daily plasma exchange, but she developed respiratory failure requiring mechanical ventilation.

A broad spectrum of virological and microbiological analyses including hantavirus diagnostics was performed. A recombinant immunoblot with antigens from hantavirus species Puumala, Dobrava-Belgrade, Seoul, Hantaan, and Sin Nombre virus (recomLine HantaPlus, Mikrogen GmbH, Neuried, Germany) was negative for hantavirus-specific IgM or IgG antibodies. However, a nested reverse transcription PCR (RT-PCR) analysis targeting the L segment of the hantavirus genome was positive in a stored blood sample from day 5. Monitoring of the viral load was performed by quantitative real-time RT-PCR (qPCR) targeting the L segment [7]. Over a period of one month, PUUV RNA was detectable in the plasma at concentrations between approximately 180,000 and 1000 copies/mL. In parallel, viral RNA was detected in other body compartments including the CSF, bronchial lavage (BAL), and urine (Figure 1a).

In a molecular phylogenetic analysis of a representative part of the L segment (Figure 2), the PUUV strain was classified into the Swabian Jura cluster from Germany [8].

A renal biopsy on day 18 showed approximately 10% acute parenchymal necrosis, severe diffuse acute tubular epithelial damage in the vital parenchymal parts, mild interstitial edema, and low-grade lymphocytic interstitial inflammation with clear predominance of CD3+ T cells (Figure 3), but no significant glomerulosclerosis, tubular atrophy, or interstitial fibrosis (approximately 10%) and no evidence for thrombotic microangiopathy, vasculitis, or glomerulonephritis. In addition, no specific glomerular or tubulointerstitial deposits of IgA, IgG, IgM, C1q, or C3c were detectable by immunohistochemistry. Thus, the findings were suggestive of mild interstitial nephritis compatible with viral infection of the kidney.

Due to clinical deterioration and persistent detection of PUUV RNA in plasma, ribavirin was administered from day 30 for 18 days with a reduced dose of 200 mg/d for patients with hemodialysis. Additionally, fresh frozen plasma (FFP) from whole blood donors with previous PUUV infection was administered for passive immunization. This plasma was derived from whole blood collected by the German Red Cross Blood Transfusion Service Baden-Württemberg—Hessen according to standard practices in agreement with the German Hemotherapy Guideline-. Active repeat blood donors with a history of PUUV infection who donated whole blood at least four months ago were identified by an announcement in social networks. The units collected and stored as regular FFP for clinical use were retrospectively identified as units donated by individuals with a history of hantavirus infection. High levels of PUUV-specific IgG antibodies could be detected in these FFP products. In total, nine convalescent FFP units (volume per unit: 190 to 390 mL) were transfused to the patient from day 36 onwards, leading to the detection of substantial amounts of PUUV-specific IgG antibodies in the patient (Figure 1b). In contrast, no anti-PUUV IgM antibodies could be detected, arguing against an endogenous antibody response of the patient. From day 39 onwards, PUUV RNA was negative in peripheral blood and all other tested body fluids (Figure 1a). Despite the antiviral treatment and viral clearance, the clinical condition of the patient did not improve and the patient died on day 81 of hospitalization.

## 3. Discussion

This lethal PUUV infection in a patient with CVID illuminates several aspects of laboratory diagnosis and potential treatment options for hantavirus disease in patients with a humoral immunodeficiency.

CVID is the most common clinically relevant primary immunodeficiency and is primarily characterized by a hypogammaglobulinemia of IgG in combination with low levels of IgA and IgM [5]. The clinical presentation shows a broad and variable phenotype including an increased risk for severe infections [5]. Prolonged virus replication and chronic disease with severe complications in CVID patients have been reported for various viruses such as herpesviruses and enteroviruses [9].

This is the first molecularly confirmed case of a fatal PUUV infection since the introduction of mandatory reporting in Germany in 2001. In general, the clinical course of PUUV infections is mild to moderate with a rapid elimination of the virus. However, this patient with severe humoral immunodeficiency developed persistent renal failure, thrombocytopenia, CNS infection with tetraparesis, and prolonged viral replication in multiple organs demonstrating the importance of the humoral immune response for the control of PUUV.

In immunocompetent patients, the laboratory diagnosis is based on serology since anti-PUUV IgM and IgG are most commonly detectable at the onset of symptoms. However, serodiagnostics failed in this case due to the patient’s impaired humoral immunity. Thus, the infection was confirmed by the presence of PUUV RNA in the peripheral blood, CSF, BAL, and urine demonstrating the importance of molecular detection of PUUV in humoral immunodeficient patients. Usually, viral RNA in PUUV-infected patients disappears from the blood within the first week after the onset of clinical symptoms [10]. In cohorts of immunocompetent patients, it has been shown that a low PUUV-specific IgG response is associated with severe disease and the presence of PUUV-specific IgG and IgA antibodies is correlated with a low PUUV RNA load [11,12]. In addition, a correlation between viral load and duration of viremia on one side and clinical severity on the other was found, albeit not statistically significant in all cases [10,12,13]. The decisive importance of the humoral immune response as well as the level and duration of RNAemia for disease severity are clearly supported by this case. Low levels of neutralizing antibodies and a high viral load have been also reported in a fatal PUUV infection in an elderly immunocompetent patient [14].

The pathogenesis of hantavirus disease is not completely understood, but it has been suggested that virus-induced immunopathological processes rather than direct cytopathogenic effects are involved in disease manifestation [15,16]. However, in the situation of the patient’s severe humoral deficiency, autoimmunity by autoantibodies can be excluded with high probability and disease manifestation can be most probably attributed to cellular immunopathogenesis which is supported by the renal histology. Severe acute tubular damage, mild interstitial edema, and mild interstitial lymphocytic infiltration were demonstrated, which are typical findings for PUUV-associated hantavirus disease [17]. However, the parenchymal necrosis was unusual and might be a histological correlate for the persistent renal failure in the patient. The interstitial infiltrate of CD3+ T cells in the absence of CD20+ B cells and the lack of detectable glomerular antibody or complement deposits make tissue damage induced by antibodies or complement activation unlikely.

The limited effectiveness of the antiviral agent ribavirin against some hantaviruses has been shown in vitro and in vivo, although trials in PUUV-infected patients with HFRS have shown conflicting results regarding the clinical efficacy of ribavirin treatment [18]. These studies [18] were conducted in immunocompetent patients, who usually show a spontaneous virus clearance within a few days, independent of therapy. In our patient, the viral load decreased from 156,000 copies/mL (prior to ribavirin) to 1800 copies/mL, but did not significantly decline further after seven days on ribavirin arguing against a strong antiviral effect of ribavirin although drug concentrations could not be monitored. Additional transfusion of FFP products from PUUV seropositive donors resulted in high anti-PUUV IgG antibody concentrations in the patient’s sera and contributed to the clearance of PUUV four days after the first administration. However, spontaneous clearance by the T-cell response, although highly unlikely after 30 days of RNAemia, cannot be excluded. After promising results of in vitro and animal studies using monoclonal antibodies or polyclonal immune sera to block hantavirus replication, the first clinical trial with hantavirus immune plasma infusion showed a decrease of CFR with borderline significance and requires further studies for confirmation [6,18].

## 4. Conclusions

This case of a fatal PUUV infection in a patient with CVID underlines the importance of the humoral response for the course of PUUV disease and illustrates the need for molecular diagnostics of PUUV in these patients. Convalescent plasma therapy should be considered in the therapy of severe hantavirus disease in patients with an impaired humoral immune response.

## Figures and Tables

**Figure 1 microorganisms-11-00283-f001:**
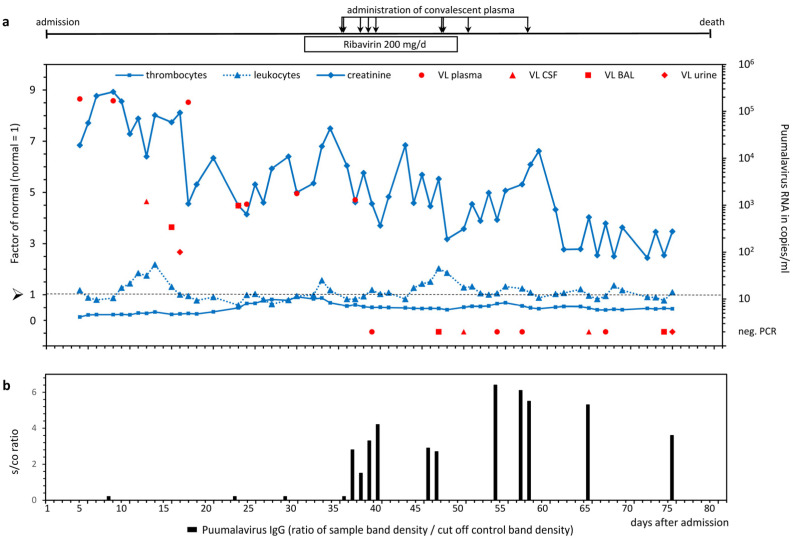
(**a**) Course of PUUV RNA concentration, and thrombocyte, leukocyte, and creatinine levels. Viral load (VL) in different compartments (cerebrospinal fluid [CSF], bronchial lavage [BAL], and urine) was quantified by real-time qPCR with a limit of detection of 100 PUUV RNA copies/mL. Values of thrombocytes, leukocytes, and creatinine in serum are depicted as factors of the normal mean value; the normal mean value of the reference range was set to 1 (dotted line) for each parameter. Treatment with ribavirin and administration of convalescent plasma are shown on top of panel (**a**). In panel (**b**), the detectable amount of anti-PUUV IgG in response to the passive immunization by PUUV immunoglobulin transfer is depicted. The four symbols between day 8 and 36 indicate negative immunoblots in the patient’s serum before administration of convalescent plasma.

**Figure 2 microorganisms-11-00283-f002:**
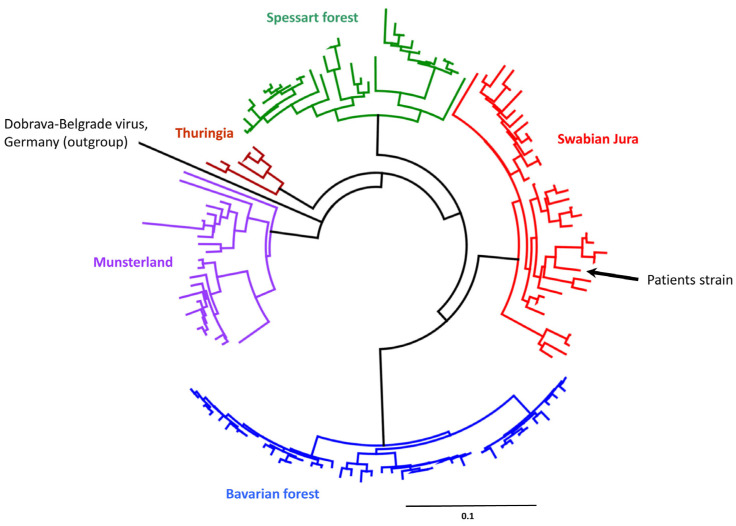
Phylogenetic analysis of partial PUUV L segment sequences in representative samples referred to us (Institute of Virology, Charité Universitätsmedizin Berlin) in our responsibility as the German National Consultant Laboratory for hantaviruses. Colors indicate phylogeographic clusters. Maximum likelihood (ML) tree based on a 345 nt alignment, calculated using the HKY85 +G +I model of nucleotide substitution. Scale bar indicates nucleotide substitutions per site. Sequence marked with an arrow stems from the patient described here. A Dobrava-Belgrade hantavirus sequence derived from an infected patient in Germany served as the outgroup.

**Figure 3 microorganisms-11-00283-f003:**
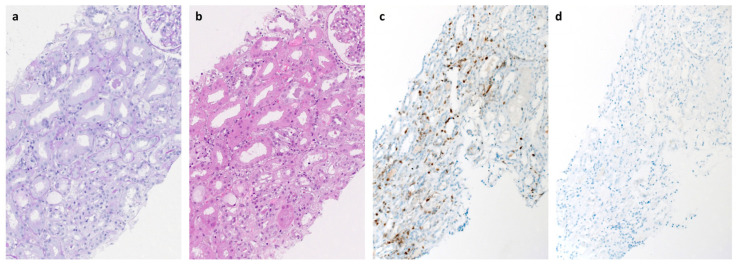
Histologic and immunohistochemical findings of the renal biopsy (day 18). Periodic acid-Schiff (PAS, **a**) and hematoxylin and eosin staining (H&E, **b**) showed severe diffuse acute tubular epithelial damage, mild interstitial edema, and low-grade focal accentuated lymphocytic interstitial inflammation. Immunohistochemical analysis showed the presence of CD3+ T cells (**c**) whereas CD20+ B cells (**d**) were not detectable.

## Data Availability

The data presented in this study are available on request from the corresponding author. The data are not publicly available due to patient’s privacy.

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
