# Peer review of "Fatal Puumala Hantavirus Infection in a Patient with Common Variable Immunodeficiency (CVID)"

_microorganisms, 2023, doi:10.3390/microorganisms11020283_

Round 1
Reviewer 1 Report
Steininger et al provide a case report of a rare fatal case of fatal Puumala virus-associated disease in a patient with common variable immunodeficiency (CVID) in Germany, even after treatment with ribavirin and fresh frozen plasma (FFP) from convalescent donors with a history of Puumala virus infection. Despite clearance of virus from the patient, clinical symptoms did not improve leading to death of the patient.
The case report is clearly and succinctly presented. The data in Figure 1 could be presented more clearly, however, this is only a minor comment. I have no further comments or criticisms.
Reviewer 2 Report
The last decade, the world has experienced an escalation of orthohantavirus infections bordering from China to India, from India to Asia to Europe and to Americas. Therefore, investigation of some factors such CVID contributing to the higher fatality rate caused by orthantaviruses is such an important study. This paper presents a case report on the fatal case caused by Puumala virus infection in a 60 year old person. The manuscript underlines the effectiveness of ribavirin and the use of convalescent plasma to treat the patient. They found that in a patient with high viral load, after introducing ribavirin, the viral load decreased from 156,000 copies/mL (prior to ribavirin) to 1,800 copies/mL. The case report can be accepted after attending to the following minor problems.
Overall Comments:
The authors should update the references to more recent ones
Lines 54-56: I would be cautious about the presenting of HFRS and HCPS in that manner, these diseases do not circulate in Europe, America and Asia as presented.
Line 61: The cited work puts the CFRs in Europe by Dobrava at 12%? It is not clear where the authors got 15 % CFRs in Europe.
Line 83: It’s not clear what is the meaning of “at last with” in this sentence
It will benefit reading if the authors could provide in the supplementary material or link/reference describing the methods used, for example to determine the viral load.
